# The direct correlation between oxidative stress and LDL-C levels in adults is maintained by the Friedewald and Martin equations, but the methylation levels in the *MTHFR* and *ADRB3* genes differ

**Jéssica Vicky Bernardo de Oliveira**[1]*, **Raquel Patrícia Ataíde Lima**[1], **Rafaella Cristhine Pordeus Luna**[1], **Alcides da Silva Diniz**[2], **Aléssio Tony Cavalcanti de Almeida**[3], **Naila Francis Paulo de Oliveira**[4], **Maria da Conceição Rodrigues Gonçalves**[1], **Roberto Texeira de Lima**[1], **Flávia Emília Leite de Lima Ferreira**[1], **Sônia Cristina Pereira de Oliveira Ramalho Diniz**[1], **Alexandre Sergio Silva**[1], **Ana Hermínia Andrade e Silva**[5], **Darlene Camati Persuhn**[4], **Maria José de Carvalho Costa**[1]

1 Postgraduate Program in Nutrition Sciences, Department of Nutrition, Health Sciences Centre, Federal University of Paraíba (Universidade Federal da Paraíba–UFPB), João Pessoa, Paraíba, Brazil,
2 Postgraduate Program in Nutrition Sciences, Department of Nutrition, Health Sciences Centre, Federal University of Pernambuco (Universidade Federal de Pernambuco—UFPE), Recife, Pernambuco, Brazil,
3 Postgraduate Program in Applied Economics, Department of Economics, Centre for Applied Social Sciences, UFPB, João Pessoa, Paraíba, Brazil, 4 Department of Molecular Biology, Centre for Exact and Natural Sciences, UFPB, João Pessoa, Paraíba, Brazil, 5 Department of Statistics, Centre for Exact and Natural Sciences, UFPB, João Pessoa, Paraíba, Brazil

* jessicavickynutri@gmail.com

## Abstract

Low-density lipoprotein (LDL-C) concentrations are a standard of care in the prevention of cardiovascular disease and are influenced by different factors. This study compared the LDL-C concentrations estimated by two different equations and determined their associations with inflammatory status, oxidative stress, anthropometric variables, food intake and DNA methylation levels in the *LPL*, *ADRB3* and *MTHFR* genes. A cross-sectional population-based study was conducted with 236 adults (median age 37.5 years) of both sexes from the municipality of João Pessoa, Paraíba, Brazil. The LDL-C concentrations were estimated according to the Friedewald and Martin equations. *LPL*, *ADRB3* and *MTHFR* gene methylation levels; malondialdehyde levels; total antioxidant capacity; ultra-sensitive C-reactive protein, alpha-1-acid glycoprotein, homocysteine, cobalamin, and folic acid levels; usual dietary intake; and epidemiological variables were also determined. For each unit increase in malondialdehyde concentration there was an increase in the LDL-C concentration from 6.25 to 10.29 mg/dL (p <0.000). Based on the Martin equation (≥70 mg/dL), there was a decrease in the DNA methylation levels in the *ADRB3* gene and an increase in the DNA methylation levels in the *MTHFR* gene (p <0.05). There was a positive relation of homocysteine and cholesterol intake on LDL-C concentrations estimated according to the Friedewald equation and of waist circumference and age based on the two estimates. It is concluded the LDL-C concentrations estimated by the Friedewald and Martin equations

**Data Availability Statement:** The database of the present study is present in a public repository. DOI 10.17605/OSF.IO/T6ZN8.

**Funding:** The authors received no specific funding for this study.

**Competing interests:** The authors have declared that no competing interests exist.

were different, and the Friedewald equation values were significantly lower than those obtained by the Martin equation. MDA was the variable that was most positively associated with the estimated LDL-C levels in all multivariate models. Significant relationships were observed based on the two estimates and occurred for most variables. The methylation levels of the *ADRB3* and *MTHFR* genes were different according to the Martin equation at low LDL-C concentrations (70 mg/dL).

## Introduction

Worldwide guidelines focus on low-density lipoprotein cholesterol (LDL-C) cut-off values as evidence-based standards for the prevention and treatment of cardiovascular disease (CVD). Many of these guidelines attribute the highest level of evidence (class 1) to LDL-C treatment goals [1–5].

In this context, the usefulness of treatment goals depends on the accuracy of the estimation used to assess LDL-C values. In this sense, a novel more accurate calculation–developed by researchers at Johns Hopkins University in Baltimore–may allow changes in management and improvement in the accuracy of LDL-C values [6].

According to the results of recent research, LDL-C continues to be routinely estimated by the Friedewald equation [1–12], which for more than 40 years has been used to support guidelines and CVD treatment management; however, it has been demonstrated that this equation tends to underestimate the LDL-C values in patients with elevated triglyceride levels (>200 mg/dL) and LDL-C below 70 mg/dL [13].

Due to the high cost of directly measuring LDL-C [13], estimating LDL-C using the Friedewald equation is still the most commonly used calculation method in clinical practice and in population-based studies. In turn, the Martin equation was applied in a study with an adult population, and its efficiency compared to the direct method (enzymatic homogeneous assay) [14] was observed. The Martin equation has also been applied in studies with individuals undergoing treatment due to high cardiovascular risk, having as a therapeutic goal the reduction in LDL-C to <70 mg/dL [6, 15, 16].

In metabolic disorders, such as vascular disease, inflammation, hypercholesterolemia and increased amounts of native LDL, changes occur in the generation of reactive oxygen species (ROS), which are responsible for causing changes in the endothelium, imbalances in oxidative metabolism and L-arginine uncoupling, causing an increase in nitric oxide release and ROS generation, in addition to endothelial dysfunction and atherogenesis [10].

MDA is the main degradation product of lipid peroxidation, and its level indirectly reflects the severity of free radical attack [17, 18]. In several studies, the influence of MDA on LDL-C was also observed in different morbidities [19–22]. The total antioxidant capacity (TAC), homocysteinemia, C-reactive protein (CRP), alpha-1-acid glycoprotein (AGP), waist circumference (WC) and cholesterol intake, based on methodological protocols for different morbidities, LDL-C levels were correlated with these variables [11, 23–27].

The relationship between LDL-C values and methylation levels, changes in the methylation profile can lead to changes in enzyme expression and fat metabolism dysregulation [28]. Additionally, external factors can regulate gene transcription and directly alter the epigenetic code, causing permanent activation of key inflammatory mediators with sustained production of pro-inflammatory mediators [29].

The DNA methylation in specific genes involved in dyslipidaemia, i.e., in the increase or decrease in LDL-C values [12, 30, 31], the following are notable: the LPL gene, which encodes lipoprotein lipase (LpL), involved in the reduced conversion of very low-density lipoprotein (VLDL) into LDL-C in plasma [32]; the beta-adrenergic receptor 3 gene (*ADRB3*), which is involved in the regulation of lipolysis and thermogenesis in white and brown adipose tissue [33]; and the gene that encodes methylenetetrahydrofolate reductase (MTHFR), which is involved in the conversion of homocysteine (Hcy) into methionine [34] and may be involved in lipid metabolism, as observed in a study of *MTHFR* polymorphism and dyslipidemia [35].

To the best of our knowledge, there are no studies comparing LDL-C values estimated using one or two methods as dependent variables and *LPL*, *ADRB3* and *MTHFR* gene methylation levels, MDA, TAC and CRP, among other independent variables, in linear regression models.

Accordingly, considering that LDL-C values and their therapeutic changes are associated with CVD prevention and treatment to a greater extent than the levels of other types of lipids [26] and that the accuracy of the estimation may help better guide the management of CVD, the present study compared the estimated LDL-C concentrations based on the equations proposed by Friedewald et al. [36] and by the new method described by Martin et al. [13] and evaluated the occurrence of an association with inflammatory status, oxidative stress, anthropometric variables, food intake and DNA methylation levels in the *LPL*, *ADRB3* and *MTHFR* genes in adults of both sexes.

## Materials and methods

### Study design

This was a cross-sectional epidemiological study associated with the project titled "Cycle II of Diagnosis and Intervention of the Food, Nutritional and Non-Communicable Diseases Statuses of the Population of the Municipality of João Pessoa/Paraíba (II Ciclo de Diagnóstico e Intervenção da Situação Alimentar, Nutricional e das Doenças não Transmissíveis mais Prevalentes da População do Município de João Pessoa/PB—II DISANDNT/Paraíba)" [37], for which data were collected between May 2015 and May 2016.

As for the inclusion criteria, individuals of both sexes were included in the sample; between 20 and 59 years of age; from different socioeconomic statuses and using medication or not. As for the exclusion criteria, the following were excluded from the original II DISANDNT / PB database: Individuals with neuro-psychiatric disorders; users of multivitamin supplements, minerals, anorectic or anabolic agents; pregnant women and infants; for the present study, individuals with a calorie intake below 700Kcal [38] and individuals with triglyceride levels above 400mg/dL were also excluded.

After the selection of the eligibility criteria, 236 individuals were selected and duly instructed as to the objectives of the study according to the ethical guidelines; the subjects agreed to participate by signing an informed consent. The research protocol of the above mentioned project, to which the present study is linked, was approved by the Research Ethics Committee of the Health Sciences Center of the Federal University of Paraíba (Universidade Federal da Paraíba, UFPB), under the protocol number 0559/2013. This study was conducted in accordance with the Declaration of Helsinki.

This information and other information about the sampling are further detailed in recently published studies [39, 40], with the exception of the levels of methylation of the LPL and MTHFR gene, the calculated values of LDL-C in accordance with Martin [13] and the stratification of the sample according to the LDL-C values.

To improve the characterization of the studied population, sex, lipid profile and gene methylation levels, distributed according to estimated LDL-C values–based on the Friedewald and Martin equations–were classified according to the Third Report of the National Cholesterol Education Program (NCEP) Expert Panel on Detection, Evaluation, and Treatment of High Cholesterol in Adults (ATP III), using the following cut-off values as a goal for LDL-C values: <130 mg/dL, corresponding to values close to the optimal value, used for individuals with low cardiovascular risk; and the treatment goal of <70 mg/dL, recommended for individuals with high cardiovascular risk [41].

## Data collection

The home visits and the application of the questionnaires were carried out by undergraduate students of the Nutrition Program and master's and PhD students of the Graduate's Program in Nutrition Sciences (PPGCN) of the UFPB. All were trained before the start of data collection and also participated in the pilot study. The questionnaires were applied to characterize epidemiological, demographic and socioeconomic, lifestyle, nutritional assessment, dietary intake and biochemical data.

## Nutritional and food intake assessment

Weight and height measurements were carried out in triplicate and the mean of the three values was used. Body Mass Index (BMI) was calculated using the weight (kg) divided by the square of the height (meters), and the cut-off points recommended for adults aged between 20 and 59 years old by the World Health Organization (WHO) [42] were used. The waist circumference (WC) was used to determine abdominal obesity, according to the American Heart Association (AHA), with the cut-off point being $\geq$ 88 cm for women and $\geq$102 cm for men [43].

To assess the regular food intake of the individuals, three 24-hour dietary recalls (24HR) were carried out, contemplating one at the weekend, with an interval of 15 days since the beginning of data collection, according to a previously-published study [40].

To fill the 24HR, a photo album with homemade measures of food was used based on the real weight of the average intake of foods validated for this population, minimizing possible biases of this method [44, 45]. The food items were evaluated and converted into calorie and nutrient quantities with the Dietwin nutrition software and the multiple sources method (MSM) was used to estimate the regular intake of individuals of repeated measurements in a given period; the variation in intake was not affected by the method [46].

## Biochemical analyses

Blood samples were collected by an experienced nurse in the individuals' homes, after a fasting period of 12 hours. The blood collection followed the standards for the use of piercing/cutting instruments. For the evaluation of the lipid profile, we used kits for the determination of total cholesterol and triglycerides of the Labtest® do Brazil brand following the Trinder enzymatic method, and for HDL cholesterol we used the Labtest® do Brazil brand following the precipitation method. PCR-us and AGP were quantified by immunoturbidimetry in serum samples; the concentrations were determined with specific commercial kits (Labtest, Minas, Brazil) according to the manufacturer's instructions. We used a previously described protocol for the analysis of antioxidant activity through plasma MDA and TAC in serum [47]. Homocysteine levels were quantified using high performance liquid chromatography (HPLC) [48]. The serum concentrations of folic acid and vitamin B12 were measured by chemiluminescence and a electrochemiluminescence immunoassay, respectively [39].

## LDL-C estimation using the Friedewald and Martin equations

The LDL-C values estimated with the Friedewald equation were calculated using the following formula: (total cholesterol)—(high-density lipoprotein cholesterol [HDL-C])—(triglycerides/5), in mg/dL [36]. The calculation of the values estimated with the Martin equation was performed using a smartphone application (LDL Cholesterol Calculator. Available at: https://www.hopkinsmedicine.org/apps/all-apps/ldl-cholesterol-calculator. Accessed on: August 2018) [6]. This new approach integrates an individualized factor in the denominator to capture the performance of triglycerides in explaining the variance in VLDL-C, where the individual factor is chosen among 180 different possible factors according to the concentrations of triglycerides and non-HDL-C [13].

## Cardiovascular risk

For the total sample, the Framingham risk score was estimated using an online calculator (Available at: (https://www.framinghamheartstudy.org/fhs-riskfunctions/cardiovascular-disease-10-year-risk/). Accessed on: May 2018), which estimates the risk of an individual to develop coronary artery disease over the next 10 years. The evaluation of this score is based on the following classification: <10% low risk; 10–20% moderate risk; and> 20% high risk.

The diagnosis of arterial hypertension and medication use were self-reported, based on a medical consultation. For non-hypertensive patients, the normal systolic pressure value (120 mmHg) was used, and for hypertensive patients, the value referring to mild (140 mmHg) and moderate (160 mmHg) systolic hypertension were used [49], respecting the heterogeneity of the population regarding the studied variables [39, 40].

The Framingham risk score was estimated to assist in the characterization of the sample regarding cardiovascular risk, but due to the lack of blood pressure values, it was not included in the regression models.

## Analysis of the methylation levels of the *MTHFR*, *ADRB3* and *LPL* genes

For the analysis, blood samples were collected at 8:30 a.m. ± 10 min from all individuals to avoid a potential sampling effect. To isolate DNA, TRIzol-LS was used according to the protocol described and adapted by Miller and Dykes [50]. For the DNA standards and participant samples, whole DNA was quantified on a NanoDrop ND-2000 UV-Vis spectrometer (Thermo Scientific, USA). Genomic DNA was modified by bisulfite using a Cells-to-CpG™ conversion kit (Applied Biosystems, Life Technologies, California) according to the manufacturer's instructions. PCR amplification and high-resolution melting (HRM) analysis were performed using an Applied Biosystems 7500 Fast System and the accompanying software. The primer sequences for each gene were obtained using MethPrimer Software (https://www.urogene.org/methprimer/) according to the promoter region sequence entered into the Genome Browser (https://genome.ucsc.edu/).

The final volume of the solution used for PCR was 20 μl, containing: AmpliTaq™ Gold 360 Buffer, 10x, 25mM Mg Chloride, MeltDoctor™ HRM Dye (20x), 25nM of each initiator containing the primer sequences for the genes, according to Fig 1 and 1 μl of bisulfite-converted genomic DNA. The qPCR consisted of an initial enzyme activation at 95˚C which lasted for 10 minutes, followed by 40 cycles, each consisting of 15 s at 95˚C, 60 s at 60˚C, 10 s at 95˚C, 60 s at 60˚C, 15 s at 95˚C, and 15 s at 60˚C the design of the studied genes are described in Box 1.

To calculate the levels of methylation, DNA standards converted to commercially available bisulfite were acquired as positive (100% Methylated) and negative control (0% Methylated) CpGenome (Chemicon, Milipore, Billerica, MA, USA) and we performed the appropriate dilutions to obtain the following ratios: 0, 25, 50, 75 and 100%; all samples were performed at

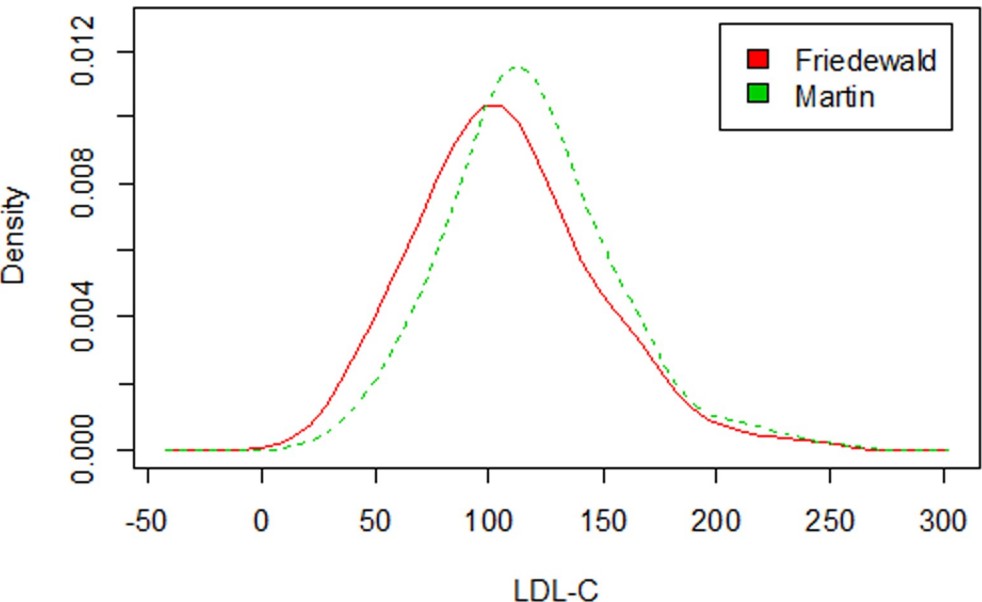

**Fig 1. LDL-C concentration density curves according to the estimation methods.**

least in triplicate to acquire the standard curves and compare with the fusion profile of the samples.

## Statistical analysis

All statistical analyses were performed with the software R, 3.3.2 [51]. Initially, the sample was characterized using measures such as simple frequency, percentage, measures of central tendency and dispersion. Next, a proportions test was used to determine whether there was a difference between the proportion of individuals with LDL-C values lower, higher and equal to 130 mg/dL and lower, higher and equal to 70 mg/dL for patients of each sex.

The data were tested for the presence of normality using the Lilliefors test, which is a derivation of the Kolmogorov-Smirnov test [52]. In addition, to determine whether there was a difference, on average, in the variables total cholesterol (TC), HDL, triglycerides, *LPL* gene methylation level, *ADRB3* gene methylation level and *MTHRF* gene methylation level according to the classes "LDL up to 130 mg/dL", "LDL equal to or greater than 130 mg/dL", "LDL up

## BOX 1. Primer design.

| Gene | Primers (F/R) | $T_A$ (°C) | Product (bp) | CpG sites | Genomic Location |
|---|---|---|---|---|---|
| *MTHFR* | F 5'–GGTCAGAAGCATATCAGTCATGAG– 3' | 58° | 213 bp | 6 | Chr1: 11,785,723– 11,805,413) |
| | R 5'–CTGGGAAGAACTCAGCGAACTCA– 3' | | | | |
| *LPL* | F 5'–GATGTTGAATATTGAATTATTTTAGTAATA– 3' | 63° | 235 bp | 4 | Chr8: 19,939,253– 19,967,259) |
| | R 5'–AAAAATCACCATACCCAACTAATTTT– 3' | | | | |
| *ADRB3* | F 5'–TAGGTGATTTGGGAGATTTTTTTT– 3' | 60° | 238 bp | 4 | Chr8: 37,962,990– 37,966,599 |
| | R 5'– CCCCTAACA ACCCACTAATATTAAC– 3' | | | | |

F–forward; R–reverse; $T_a$–annealing temperature.

to 70 mg/dL", and "LDL equal to or greater than 70 mg/dL", estimated by the Friedewald and Martin equations, the t-test, for comparisons of means, and the Mann-Whitney test, for comparison of medians, when the size of the subsample was small, were used.

To determine if there were significant relationships among the LDL-C values estimated by the Friedewald and Martin equations and the methylation levels of the *LPL*, *MTHFR and ADRB3* genes and the other study variables, the linear regression models described below were proposed:

Model 1: $\mu$(LDL Friedewald) = $\beta_0$ + $\beta_1$*LPL* gene methylation levels + $\beta_2$*MDA + $\beta_3$*TAC + $\beta_4$*PCR + $\beta_5$*alpha-1-glyco + $\beta_6$*Hcy + $\beta_7$*B12 + $\beta_8$*folate;

Model 3: $\mu$(LDL Friedewald) = $\beta_0$ + $\beta_1$* *LPL* gene methylation levels + $\beta_2$*CHO + $\beta_3$*PTN + $\beta_4$*LIP + $\beta_5$*vitamin A + $\beta_6$*mono + $\beta_7$*B12 + $\beta_8$*oleic acid + $\beta_9$*omega 3 + $\beta_{10}$*omega 6 + $\beta_{11}$*saturated fat + $\beta_{12}$*cholesterol + $\beta_{13}$*polyunsaturated fats + $\beta_{14}$*trans fats + $\beta_{15}$*vitamin C + $\beta_{16}$*vitamin E + $\beta_{17}$*betacarotene + $\beta_{18}$*selenium + $\beta_{19}$*folate;

Model 5: $\mu$(LDL Friedewald) = $\beta_0$ + $\beta_1$*LPL* gene methylation levels + $\beta_2$*sex + $\beta_3$*age + $\beta_4$*income + $\beta_5$*physical activity + $\beta_6$*smoking + $\beta_7$*alcohol consumption + $\beta_8$*WC + $\beta_9$*hypolipidaemic drugs + $\beta_{10}$*omega 6 + $\beta_{11}$*morbidities related to CVDs.

The same models proposed for the Friedewald equation were also proposed for the Martin equation (models 2,4,6) and for the *ADRB3* and *MTHRF* genes. For all analyses described above, the level of significance adopted was 5%.

## Results

The study consisted of 236 adult individuals, of which 158 (67%) were female and 78 were male (33%), with a median age of 37.5 years.

There was a significant difference (p = 0.003) between mean LDL-C concentrations in the total sample estimated by the Friedewald and Martin equations, with LDL-C concentrations of 107.24 mg/dL and 117.81 mg/dL, respectively.

Of the 236 individuals included in the present study, 83% and 92% presented LDL-C $\geq$ 70 mg/dL, and 25% and 33% showed LDL-C $\geq$ 130 mg/dL according to Friedewald and Martin equations, respectively.

Fig 1 shows that the LDL-C density curve calculated by the Friedewald equation is slightly more dispersed than that calculated by the Martin method, which can also be observed by the estimated standard deviation, which was 39.31 for LDL-C (Friedewald) and 36.49 for LDL-C (Martin). According to the non-parametric Mann-Whitney test, there was a significant difference between LDL-C values according to the estimation method, at the 1% significance level (W = 23062, p-value = 0.001).

Table 1 shows that there was a difference according to sex in the values estimated between the intervals less than 130 mg/dL and above 130 mg/dL; more women had LDL-C levels lower than 130 mg/dL. There was also a difference between the intervals presented above for the mean TC and triglyceride values.

As for the WC values (male and female) distributed according to the LDL-C values calculated according to Friedewald, there was a difference between the values < and $\geq$ 130 mg/dL in relation to the values < and $\geq$ 70. It was observed that individuals who had a higher WC, also presented higher values of LDL-C.

As for the WC values for the male sex distributed according to the LDL-C values calculated according to Martin, there also was a difference between the values < and $\geq$ 130 mg/dL in relation to the values < and $\geq$ 70.

**Table 1. Characteristics of the sample regarding sex, waist circumference, lipid profile and methylation levels of the *LPL*, *ADRB3* and *MTHFR* genes according to different LDL-C estimates and cut-off values.**

| Lipid profile | Mean | SD | LDL-C F (mg/dL) <130 <70 | LDL-C F (mg/dL) ≥130 ≥70 | P-value | LDL-C M (mg/dL) <130 <70 | LDL-C M (mg/dL) ≥130 ≥70 | P-value |
|---|---|---|---|---|---|---|---|---|
| Female | - | - | 117 (50%) | 41 (17%) | <0.000*[a] | 105 (45%) | 53 (22%) | 0.000*[a] |
|  |  |  | 24 (10%) | 134 (56.8) | <0.000*[a] | 13 (5%) | 145 (61%) | 0.000*[a] |
| Male | - | - | 61 (26%) | 17 (7%) | <0.000*[a] | 55 (23%) | 23 (10%) | <0.000*[a] |
|  |  |  | 17 (7%) | 61 (25.8%) | <0.000*[a] | 6 (2%) | 72 (30%) | <0.000*[b] |
| Female waist size (cm) | 84.44 | 14.8 | 82.84 ± 14.94 | 88.61 ± 13.81 | 0.019*[c] | 82.48 ± 15.56 | 88.22 ± 12.58 | 0.010*[c] |
|  |  |  | 81.08 ± 16.08 | 84.99 ± 14.59 | 0.250[c] | 74.79 ± 12.87 | 85.31 ± 14.72 | 0.008*[c] |
| Male waist size (cm) | 87.18 | 14.6 | 86.39 ± 15.52 | 90.03 ± 10.25 | 0.259[b] | 85.13 ± 14.65 | 92.08 ± 13.40 | 0.048*[b] |
|  |  |  | 83.80 ± 14.13 | 88.05 ± 14.65 | 0.298[b] | 79.17 ± 13.25 | 87.85 ± 14.54 | 0.177[b] |
| Total cholesterol (mg/dL) | 186.4 | 42.1 | 169.98 ± 28.32 | 236.60 ± 37.30 | <0.000*[b] | 164.39 ± 24.69 | 232.60 ± 32.40 | <0.000*[b] |
|  |  |  | 139.29 ± 24.26 | 196.25 ± 38.13 | <0.000*[b] | 121.84 ± 13.49 | 192.00 ± 38.86 | <0.000*[b] |
| HDL (mg/dL) | 43.78 | 11 | 43.45 ± 11.19 | 44.78 ± 10.43 | 0.413[b] | 43.36 ± 11.62 | 44.67 ± 9.59 | 0.360[b] |
|  |  |  | 40.90 ± 13.67 | 44.38 ± 10.29 | 0.129[b] | 46.37 ± 13.71 | 43.55 ± 10.74 | 0.394[b] |
| Triglycerides (mg/dL) | 138.8 | 73.7 | 132.12 ± 69.19 | 159.38 ± 83.43 | 0.027*[b] | 129.33 ± 70.12 | 158.80 ± 77.47 | 0.124[b] |
|  |  |  | 142.37 ± 85.15 | 138.07 ± 71.29 | 0.764[b] | 115.89 ± 64.34 | 140.83 ± 74.26 | 0.124 [b] |
| *LPL* gene methylation level (%) | 36% | 19% | 36% ± 20% | 36% ± 17% | 0.901[b] | 36% ± 19% | 37% ± 18% | 0.632[b] |
|  |  |  | 34% ± 18% | 38% ± 19% | 0.430 [b] | 37% ± 23% | 36% ± 19% | 0.919 [b] |
| *ADRB3* gene methylation level (%) | 41% | 18% | 42% ± 18% | 38% ± 17% | 0.098[b] | 43% ± 18% | 39% ± 18% | 0.151[b] |
|  |  |  | 43% ± 19% | 41% ± 18% | 0.452 [b] | 52% ± 19% | 40% ± 18% | 0.010*[c] |
| *MTHRF* gene methylation level (%) | 35% | 18% | 34% ± 18% | 36% ± 17% | 0.487[b] | 34% ± 18% | 36% ± 18 | 0.381[b] |
|  |  |  | 31% ± 16% | 36% ± 18% | 0.165 [b] | 26% ± 10% | 36% ± 18% | 0.031*[c] |

a: proportions test

b: t-test

*: significant results

c: Mann-Whitney test; -: the calculation does not apply or does not make sense; F: Friedewald; M: Martin; LDL-C: low-density lipoprotein cholesterol.

In the total sample, 14% of individuals were hypertensive, and after calculating the Framingham score–using a hypothetical value of 160 mmHg (moderate hypertension)– 82% presented mild risk, 13% presented moderate risk, and 5% presented high risk. However, when the Framingham score was calculated using 140 mmHg (mild hypertension), 89% presented mild risk, 8% presented moderate risk, and 3% presented high risk. When the corresponding normal blood pressure value of 120 mmHg was applied, excluding the self-reported hypertensive patients from the total sample, 93% of the individuals presented mild risk, 6% presented moderate risk, and 0.5% presented high risk.

There were differences in the distribution between the methylation levels of the *ADRB3* gene (p = 0.010) and between the methylation levels of the *MTHFR* gene (p = 0.031) according to the Mann-Whitney test for values classified according to the cut-off value of 70 mg/dL (Table 1).

As described in Table 2, using the methylation levels of the *LPL* and *ADRB3* genes as independent variables, the variables MDA and Hcy positively influenced the LDL-C value estimated with the Friedewald equation. The LDL-C Friedewald increased, on average, 10.06 mg/dL for each nmol/L MDA; and the LDL-C Friedewald increased, on average, 0.55 mg/dL for each µmol/L Hcy. Regarding the Martin equation with these same two genes, the MDA variable also positively influenced the variable LDL-C, and for each nmol/L MDA and µmol/L

**Table 2. Relationship between LDL-C levels estimated based on the Friedewald and Martin equations and the methylation levels of the *LPL*, *ADRB3* and *MTHFR* genes, oxidative stress, and plasma levels of vitamin B12 and folate.**

Model 1—Response variable: LDL-C Friedewald equation

| Variable | *LPL* coefficient | p-value | *ADRB3* coefficient | p-value | *MTHFR* coefficient | p-value |
|---|---|---|---|---|---|---|
| Intercept | 75.13 | >0.000 | 75.13 | >0.000 | 72.99 | >0.000 |
| Gene (%) | 11.10 | 0.408 | 11.10 | 0.426 | 19.17 | 0.201 |
| MDA (nmol/L) | 10.06 | 0.000 * | 10.06 | 0.000* | 10.29 | 0.000* |
| TAC (%) | 10.17 | 0.566 | 10.17 | 0.55 | 7.02 | 0.693 |
| CRP (mg/L) | -1.10 | 0.143 | -1.10 | 0.117 | -1.18 | 0.115 |
| AGP (mg/dL) | 0.03 | 0.814 | 0.03 | 0.875 | 0.02 | 0.903 |
| Hcy (µmol/L) | 0.55 | 0.040* | 0.55 | 0.040* | 0.56 | 0.038* |
| Vitamin B12 (pg/ml) | -0.03 | 0.145 | -0.03 | 0.152 | -0.03 | 0.153 |
| Folic Acid (ng/ml) | -0.01 | 0.599 | -0.01 | 0.531 | -0.00 | 0.962 |
| Model 2—Response variable: LDL-C Martin Equation | | | | | | |
| Intercept | 100.20 | >0.000 | 109.70 | >0.000 | 96.48 | >0.000 |
| Gene (%) | 6.61 | 0.604 | -15.52 | 0.247 | 18.04 | 0.205 |
| MDA (nmol/L) | 6.37 | 0.020* | 6.25 | 0.022* | 6.57 | 0.016* |
| TAC (%) | 1.14 | 0.946 | 2.4 | 0.887 | -1.53 | 0.928 |
| CRP (mg/L) | -0.86 | 0.230 | -0.924 | 0.195 | -0.91 | 0.199 |
| AGP (mg/dL) | 0.09 | 0.487 | 0.09 | 0.508 | 0.08 | 0.535 |
| Hcy (µmol/L) | -0.12 | 0.638 | -0.11 | 0.663 | -0.11 | 0.665 |
| Vitamin B12 (pg/ml) | -0.02 | 0.284 | -0.02 | 0.278 | -0.02 | 0.286 |
| Folic Acid (ng/ml) | 0.00 | 0.973 | -0.00 | 0.899 | 0.01 | 0.637 |

*: $p < 0.005$; LDL-C: low-density lipoprotein cholesterol; AGP: alpha-1-acid glycoprotein; Hcy: homocysteine; MDA: malondialdehyde; CRP: C-reactive protein.

Hcy, the LDL-C estimated by the Martin equation increased, on average, 6.37 mg/dL and 6.25 mg/dL, respectively.

With the methylation levels of the *MTHFR* gene as one of the independent variables, for each nmol/L MDA, the LDL-C Friedewald increased, on average, 10.29 mg/dL, and for each µmol/L Hcy, the LDL-C Friedewald increased, on average, 0.56 mg/dL. According to the Martin equation with this same gene, the variable MDA had a positive influence on the variable LDL-C, and for each nmol/L MDA, LDL-C increased, on average, 6.57 mg/dL.

There was a positive influence of cholesterol intake on the LDL-C concentrations calculated according to the Friedewald equation, based on Model 3, which has the methylation levels of the *LPL*, *ADRB3* and *MTHFR* genes as independent variables (Table 3).

**Table 3. Relationship between LDL-C levels estimated with the Friedewald and Martin methods and the methylation levels of the *LPL*, *ADRB3*, *MTHFR* genes and habitual dietary intake.**

| Model 3—Response variable: LDL-C Friedewald | | | | |
|---|---|---|---|---|
| Variable | Coefficient | LB-UB (95%) | Z statistic | p-value |
| Intercept | 109.10 | 98.16 ± 120.04 | 9.97 | >0.000 |
| *LPL* gene (%) | 10.85 | -2.83 ± 24.53 | 0.79 | 0.428 |
| *ADRB3* gene (%) | -8.93 | -23.73 ± 5.87 | -0.60 | 0.547 |
| *MTHFR* gene (%) | 10.02 | -5.42 ± 25.46 | 0.65 | 0.517 |
| Cholesterol (mg) | 0.03 | 0.02 ± 0.05 | 2.15 | 0.033* |

Model adjusted for the intake of the nutrients were included in the explanatory model; *: $p < 0.005$; LDL- C: low-density lipoprotein cholesterol; LB: lower boundary; UB: Upper boundary. No significant relationship was found in this model for Martin.

**Table 4. Relationship among LDL-C levels estimated based on the Friedewald and Martin methods and methylation levels of the *LPL*, *ADRB3*, and *MTHFR* genes and demographic, epidemiological, lifestyle and anthropometric variables.**

| Model 6—Response LDL-C Martin | | | | |
|---|---|---|---|---|
| Variable | Coefficient | LB-UB (95%) | Z statistic | p-value |
| Intercept | 26.70 | 7.63 ± 45.77 | 1.40 | 0.163 |
| *LPL* gene | 3.16 | -8.61 ± 14.93 | 0.27 | 0.789 |
| *ADRB3* gene | -9.42 | -22.14 ± 3.30 | -0.74 | 0.460 |
| *MTHFR* gene | 10.58 | -2.06 ± 23.22 | 0.84 | 0.404 |
| Age | 0.80 | 0.59 ± 1.01 | 3.86 | 0.000* |
| Waist circumference | 0.51 | 0.35 ± 0.67 | 3.21 | 0.001* |
| Model 5—Response variable: LDL-C Friedewald | | | | |
| Intercept | 42.35 | 19.96 ± 64.74 | 1.89 | 0.060 |
| *LPL* gene | 13.02 | -7.92 ± 18.87 | 0.90 | 0.367 |
| *ADRB3* gene | -1.42 | -15.92 ± 13.08 | -0.10 | 0.922 |
| *MTHFR* gene | 14.00 | -0.39 ± 28.39 | 0.97 | 0.331 |
| Age | 0.50 | 0.26 ± 0.73 | 2.11 | 0.036* |
| Waist circumference | 0.42 | 0.24 ± 0.60 | 2.33 | 0.020* |

Model adjusted for demographic, epidemiological, lifestyle and anthropometric variables included in the explanatory model

*: $p < 0.005$; LDL-C: low-density lipoprotein cholesterol; LB: lower boundary; UB: Upper boundary.

Additionally, there was a positive relationship among age and WC with LDL-C concentrations based on the two LDL-C estimates using Models 5 and 6; in contrast, for Model 5, there was not a positive relationship between the methylation level of the *LPL* gene and cholesterol and LDL-C levels calculated based on the Friedewald equation (Table 4).

## Discussion

LDL-C concentrations varied as a function of the equations used to estimate their values, both for the Friedewald equation and for the Martin equation, and were significantly different. In this sense, the present population-based study identified that there is a distinction in the dimension of the problem; that is, LDL-C concentrations vary according to the estimation method used.

The inaccuracies of the formula with regard to triglyceride levels > 400 mg/dL (levels observed in 1% of the population of the present study) were recognized by Friedewald, Levy and Fredrickson [36]. However, even when triglyceride levels are below 400 mg/dL, researchers have suggested that the LDL-C calculated by the Friedewald formula underestimates LDL-C concentrations, corroborating the data found in the present study, where the mean LDL-C of the total sample was 107.24 and 117.81 mg/dL according to the Friedewald and Martin equations, respectively. The LDC-C concentrations estimated according to the Friedewald equation can therefore erroneously classify the CVD risk, particularly in individuals with high triglyceride levels (≥200 mg/dL; levels observed in 18% of the population in the present study, who presented mild cardiovascular risk) [6, 13].

The results from the only study found in the literature that compared the LDL-C levels estimated using the two equations in an adult population cannot be compared with the results of the present study because the objectives were different, considering that no comparisons were made between the two equations but, rather, between their results and the direct method [14]. However, notably, Lee et al. [14] observed that the Martin equation exhibited significantly higher overall agreement with the NCEP-ATP III guideline classifications than did the Friedewald equation, with values of 82.0% and 78.2%, respectively.

In the multivariate regression analysis, MDA was the only variable that positively correlated with LDL-C concentrations, calculated based on the Friedewald [36] and Martin et al. [13] equations, in all multivariate models analysed and composed of independent variables such as methylation levels of the *LPL*, *ADRB3* and *MTHFR* genes, oxidative stress, inflammation, abdominal obesity, food intake and epidemiological variables. In addition, a positive relationship was found among Hcy and cholesterol with the LDL-C concentrations estimated according to the Friedewald equation in all models.

There was also a relationship among age and WC with LDL-C concentrations, estimated based on the two equations, with the exception of the model that included the methylation levels of the *LPL* gene as independent variable and LDL-C estimated based on the Friedewald equation as the dependent variable [36].

This study is unprecedented with regard to the identification of relationships and analysis of the influence of variables such as *LPL*, *ADRB3* and *MTHFR* gene methylation, oxidative stress, inflammation, abdominal obesity, food intake and epidemiological variables on the LDL-C concentrations estimated by different equations.

In the present study, 50% of the individuals presented no morbidity, 17% had more than one morbidity, and 33% had one morbidity, and most individuals showed methylation levels between 35% and 41%. Differences in the methylation levels of the *MTHFR* and *ADRB3* genes were identified only with the LDL-C concentrations estimated according to the Martin equation and at the cut-off value of 70 mg/dL. Although in the regression there was no significant relationship, we observed that at lower concentrations of LDL-C, methylation levels were statistically reduced when compared to higher values of LDL-C. Regarding the *ADRB3* gene, in the present study *ADRB3* methylation levels were lower at LDL-C levels $\geq$ 70 mg/dL. Although the population is different from the present study, it has been shown in men with familial hypercholesterolemia (FH) that the mean methylation levels of the *ADRB3* gene were lower and correlated with high levels of LDL-C [30]. Interestingly, those authors found that hypermethylation was associated with higher *ADRB3* mRNA levels. They also detected that severely obese men also showed an overall *ADRB3* hypomethylation compared to FH men, suggesting that *ADRB3* hypomethylation might be associated with obesity, body fat distribution and related disorders in men. The β-3 adrenergic receptor (β-3AR), a protein polypeptide consisting of 402 amino acids is, a member of the G protein-coupled receptor superfamily and mainly mediates the decomposition and heat generation of fat [53]. If the hypomethylation found in the present study could confirm its relationship with the decrease in *ADRB3* transcript level, the higher LDL-C levels found could be better understood.

In contrast, another study showed that in women with excess weight, higher intake of monounsaturated and polyunsaturated fats from hazelnut oil was associated with lower *ADRB3* methylation levels and greater reductions of LDL-C levels [39].

This result was inverse for the *MTHFR* gene, that is, at LDL-C levels $\geq$ 70 mg/dL, *MTHFR* methylation levels were higher. In the reviewed literature, hypermethylation of the *MTHFR* gene was associated with higher levels of TC and LDL-C in diabetic individuals [31]. MTHFR is an enzyme that acts on the folate cycle, and in MTHFR-deficient mice it was shown that the deficiency affects apolipoprotein levels and leads to lipid deposition [54]. In addition, the relationship between hypermethylation and decreased MTHFR transcript levels has already been shown [55]. Thus, lower MTHFR levels could lead to increased lipid levels. Although the mechanism of this enhancement is not yet clear, some studies of *MTHFR* polymorphisms that lead to decreased enzyme activity also show an association with higher lipid levels [31, 56].

Regarding the levels of methylation of the *LPL* gene, no relationship was found in the present study between the methylation levels of this gene and LDL-C concentrations, corroborating studies based on the analysis of the visceral adipose tissue of individuals with metabolic

syndrome [12], of leukocytes from individuals with familial hypercholesterolemia [57] and of the placenta of pregnant women with gestational diabetes mellitus [58], thus also in individuals with defined clinical diagnoses.

Malondialdehyde (MDA) is an end-product of the radical-initiated oxidative decomposition of polyunsaturated fatty acids; therefore, it is frequently used as a biomarker of oxidative stress [23]. In the present study, 12% of the total population had high MDA, and 25% and 33% had high LDL-C (> 130 mg/dL), according to the Friedewald and Martin equations, respectively. Thus, although less than 15% of the population presented high MDA values, these values likely affect the lipid profile, increasing the LDL-C concentrations in the analysed models in the study population, with 75% to 67% of the LDL-C concentrations considered adequate, according to the two equations used.

In the Veterans Affairs Diabetes Trial, which investigated the association between MDA and diabetes mellitus, it was observed that the levels of MDA and LDL-C in circulating immune complexes can predict the occurrence of myocardial infarction, in addition to acute cardiovascular events in patients with type 2 diabetes [59]. However, the precise mechanism by which MDA can cause endothelial cytotoxicity is not known [23].

Regarding the relationship between MDA and LDL-C, several studies have shown that MDA and LDL-C values decrease in animal models with interventions based on the intake of different foods: Garlic (*Allium sativum* L.) polysaccharide [21], Chicory (*Cichorium intybus* L.) polysaccharides [22], *Morechella esculenta* polysaccharide [27].

Regarding the other study variables, considering that vascular inflammation is a fundamental mechanism in the progression of atherosclerosis and acute coronary syndromes [60], CRP stands out as an inflammation biomarker and is recommended as a secondary risk assessment factor in treatment decision-making [61, 62], particularly with regard to the treatment of high blood cholesterol [63, 64]. However, possibly because the mean CRP values for the majority of the population (81%) remained within the reference values, this variable did not influence LDL-C levels.

Hcy was included as a variable because increased Hcy affects the vascular endothelium, promoting structural endothelial injury [65]. Several mechanisms have been proposed to explain the association between Hcy and coronary artery disease, including the stimulation of LDL-C oxidation [66, 67]. In the present study, 14% of individuals had Hcy values above the reference range; however, the increased values may not have been sufficient to significantly influence the increase in LDL-C levels.

Regarding triglyceride values, Ference et al. [68] emphasized that the clinical benefit of reducing triglyceride levels is directly similar to the clinical benefit of reducing LDL-C levels per unit difference in apolipoprotein B; reduced triglyceride and LDL-C levels are associated with a lower risk of CVD. In the present study, triglyceride values increased significantly as a function of increased LDL-C. Further studies are needed to explain why there was only a difference between the triglyceride values when distributed at cut-off values of <130 mg/dL and ≥ 130 mg/dL LDL-C estimated according to the Friedewald equation.

Regarding the relationship among demographic, anthropometric and epidemiological variables, there was a positive influence of age, habitual cholesterol intake and WC on LDL-C levels. In the literature, advanced age and male sex have long been well established risk factors for CVD [69–72].

With regard to cholesterol intake, [73] observed that the intake of a high-fat diet (70.8%) increased the LDL-C levels in an experimental group; however, in the control group (29% fat), there was no significant difference. In the present study, even when the total fat intake was within the reference range (29%) [70] and the habitual cholesterol intake was 243.51 mg, an influence of this intake on LDL-C levels was identified.

Saturated fat, which is part of the habitual intake of this population, is a good source of atherogenic saturated fatty acids and is found in products such as coconut oil, which is rich in lauric fatty acids, and butter, which is a source of myristic and palmitic fatty acids [49].

Another explanation for this discrepancy may involve the type of fat consumed, since, in the case of the present study, the consumption of saturated fatty acids was 12%, exceeding the recommendation according to the National Institute of Heart, Lung and Blood: Detection, assessment and treatment of high blood cholesterol levels in adults (adult treatment panel III), final report, US Department of Health and Human Services, NIH Publication No. 02–5215, Bethesda, Maryland, September 2002.

WC is associated with CVD [74], and understanding changes in LDL-C with increased WC may help predict reductions in LDL-C, a result that can be achieved through weight loss. Regarding explanations for the relationship between LDL-C and WC, there are several hypotheses, which are different for values above or below the inflexion points for WC found in the population according to Laclaustra et al. [11].

In the present study, WC was not analysed by inflexion points, as in the aforementioned study, thus precluding comparisons with it. In this sense, we can only infer that according to the variable sex, the population of the present study was rated between 85% and 59% below and 14% and 39% above the WC reference values for males and females, respectively.

As limitations we emphasize that the results of this observational study, due to its innovative nature, should be confirmed in intervention or longitudinal follow-up studies to check if these associations are definitely causal or merely correlative. Another limitation is the fact that the LDL-C values were not assessed by the direct method. In addition, we emphasize the lack of pressure values, making it relevant to mention that, among the five most important studies conducted in the last decade in Brazil, two of them used self-reported data to calculate the prevalence of hypertension [49]. The strengths include the unprecedented aspect regarding the identification and analysis of the relationships between the LDL-C concentrations estimated based on different equations and the inclusion of the methylation levels of the *LPL*, *ADRB3* and *MTHFR* genes as independent variables.

In addition to adding to our understanding in relation to the Martin and Friedewald equations, our data contribute to the elucidation of molecular mechanisms involved in cellular metabolism. The molecular data can be tabulated collectively in a gene panel indicating the susceptibility to a variety of pathologies and it can be used as a tool for personalized medicine, an action that will help in diagnosis, prognosis and treatment.

## Conclusion

The LDL-C concentrations estimated by the Friedewald and Martin equations were different, and the Friedewald equation values were significantly lower than those obtained by the Martin equation.

MDA was the variable that was most positively associated with the estimated LDL-C levels in all multivariate models composed of variables associated with *LPL*, *ADRB3* and *MTHFR* gene methylation levels, oxidative stress, inflammation, abdominal obesity, food intake and epidemiological variables.

Regarding the association of the variables age and WC with the LDL-C concentrations estimated based on the two equations, a positive association was observed for all models, with the exception of the model that had the methylation levels of the *LPL* gene and LDL-C concentrations estimated by the Friedewald equation. Based only on the Friedewald equation, Hcy and cholesterol intake were two positive associations with LDL-C levels.

The methylation levels of the *ADRB3* and *MTHFR* genes were different with the Martin equation using the lowest LDL-C concentrations as cut-off values.

Further studies are needed to explain why some associations only occurred based on the LDL-C concentrations estimated by the Friedewald equation and for the methylation levels, which were only different when compared to the LDL-C concentrations at the lowest cut-off values based on the Martin equation.

## Supporting information

**S1 Table. Concentrations of LDL-C in individuals with high cardiovascular risk based on an estimated arterial pressure of 160 mmHg (5%).**
(DOCX)

**S2 Table. Concentrations of LDL-C in individuals with high cardiovascular risk based on an estimated arterial pressure of 140 mmHg (3%).**
(DOCX)

**S1 File.**
(DOCX)

## Acknowledgments

We thank everyone who contributed to the article and the project coordinator Professor doctor Maria José de Carvalho Costa.

## Author Contributions

**Conceptualization:** Jéssica Vicky Bernardo de Oliveira, Raquel Patrícia Ataíde Lima, Naila Francis Paulo de Oliveira, Maria da Conceição Rodrigues Gonçalves, Sônia Cristina Pereira de Oliveira Ramalho Diniz, Darlene Camati Persuhn, Maria José de Carvalho Costa.

**Formal analysis:** Jéssica Vicky Bernardo de Oliveira, Aléssio Tony Cavalcanti de Almeida, Roberto Texeira de Lima, Alexandre Sergio Silva, Maria José de Carvalho Costa.

**Funding acquisition:** Jéssica Vicky Bernardo de Oliveira, Aléssio Tony Cavalcanti de Almeida, Flávia Emília Leite de Lima Ferreira.

**Investigation:** Jéssica Vicky Bernardo de Oliveira, Raquel Patrícia Ataíde Lima, Maria José de Carvalho Costa.

**Methodology:** Jéssica Vicky Bernardo de Oliveira, Raquel Patrícia Ataíde Lima, Rafaella Cristhine Pordeus Luna, Alcides da Silva Diniz, Aléssio Tony Cavalcanti de Almeida, Flávia Emília Leite de Lima Ferreira, Ana Hermínia Andrade e Silva.

**Project administration:** Rafaella Cristhine Pordeus Luna, Alcides da Silva Diniz, Darlene Camati Persuhn, Maria José de Carvalho Costa.

**Resources:** Rafaella Cristhine Pordeus Luna, Naila Francis Paulo de Oliveira, Roberto Texeira de Lima, Flávia Emília Leite de Lima Ferreira, Sônia Cristina Pereira de Oliveira Ramalho Diniz, Alexandre Sergio Silva.

**Software:** Flávia Emília Leite de Lima Ferreira, Alexandre Sergio Silva.

**Supervision:** Alcides da Silva Diniz, Roberto Texeira de Lima, Sônia Cristina Pereira de Oliveira Ramalho Diniz, Ana Hermínia Andrade e Silva, Maria José de Carvalho Costa.

**Validation:** Ana Hermínia Andrade e Silva.

**Visualization:** Maria da Conceição Rodrigues Gonçalves.

**Writing – original draft:** Jéssica Vicky Bernardo de Oliveira, Raquel Patrícia Ataíde Lima, Maria da Conceição Rodrigues Gonçalves, Maria José de Carvalho Costa.

**Writing – review & editing:** Jéssica Vicky Bernardo de Oliveira, Raquel Patrícia Ataíde Lima, Maria da Conceição Rodrigues Gonçalves.

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
