## [Decision Letter · Decision Letter 0]

30 Apr 2020

PONE-D-20-07249

The direct correlation between oxidative stress and LDL-C levels in adults is maintained by the Friedewald and Martin equations, but the methylation levels in the MTHFR and ADRB3 genes differ

PLOS ONE

Dear Mrs. Oliveira,

Thank you for submitting your manuscript to PLOS ONE. After careful consideration, we feel that it has merit but does not fully meet PLOS ONE’s publication criteria as it currently stands. Therefore, we invite you to submit a revised version of the manuscript that addresses the points raised during the review process.

We would appreciate receiving your revised manuscript by Jun 14 2020 11:59PM. To enhance the reproducibility of your results, we recommend that if applicable you deposit your laboratory protocols in protocols.io, where a protocol can be assigned its own identifier (DOI) such that it can be cited independently in the future. For instructions see: http://journals.plos.org/plosone/s/submission-guidelines#loc-laboratory-protocols

We look forward to receiving your revised manuscript.

Kind regards,

Juan J Loor

Academic Editor

PLOS ONE

Journal Requirements:

2. Our internal editors have evaluated your manuscript and determined that it is within the scope of our 'Primary and Secondary Prevention of Cardiovascular Disease' Call for Papers. This collection of papers is headed by a team of Guest Editors for PLOS ONE and will encompass a diverse range of research articles. Additional information can be found on our announcement page: (https://collections.plos.org/s/prevention-cardiovascular). If you would like your manuscript to be considered for this collection, please let us know in your cover letter and we will ensure that your paper is treated as if you were responding to this call. If you would prefer to remove your manuscript from collection consideration, please specify this in the cover letter.

'We thank the funding agencies that made the project possible: Brazilian National Council for

Scientific and Technological Development (Conselho Nacional Desenvolvimento Científico e

Tecnológico - CNPq), Ministry of Health, and Research Support Foundation of the State of

Paraíba (Fundação de Amparo à Pesquisa do Estado da Paraíba - FAPESQ); and the research

collaborators (Graduate Program in Nutritional Sciences, University of Paraíba).'

'NO - Include this sentence at the end of your statement: The funders had no role in

study design, data collection and analysis, decision to publish, or preparation of the

manuscript.'

Additional Editor Comments (if provided):

Reviewers' comments:

Reviewer's Responses to Questions

**Comments to the Author**

1. Is the manuscript technically sound, and do the data support the conclusions?

Reviewer #1: Partly

2. Has the statistical analysis been performed appropriately and rigorously? 

Reviewer #1: N/A

3. Have the authors made all data underlying the findings in their manuscript fully available?

Reviewer #1: Yes

4. Is the manuscript presented in an intelligible fashion and written in standard English?

Reviewer #1: Yes

5. Review Comments to the Author

Reviewer #1: The paper “The direct correlation between oxidative stress and LDL-C levels in adults is maintained by the Friedewald and Martin equations, but the methylation levels in the MTHFR and ADRB3 genes differ” by Jéssica Vicky Bernardo de Oliveira et al. is of some interest but I have some doubts which are difficult to be showed because the text is without pages and lines number. Therefore, I have showed the words-sentences and some comments have been done.

The number of pages is considering the whole document I received (sorry, but next time I shall not make any referee work on similar situation).

Introduction:

- Pag. 8, the sentence: “The biochemical, anthropometric and consumption variables are justified based on the following description:“ does not seem to be complete.

Materials and Methods

- Pag. 11, …” Weight and height measurements were carried out in triplicate…” did they occurred at the 3 times of dietary recalls? Please to explain;

- Pag. 13, it is suggested: “blood samples were collected at 8:30 a.m. ± 10 min from all individuals to avoid a potential sampling effect.” Please to explain the reason.

Results

- Pag. 18, Table 2 I cannot imagine that Homocysteine could be “mol/L”

- Pag. 19, Table 3 only dietary cholesterol has been included? See some notes in Portuguese

- Pag. 20, Table 4 the LPL gene was not included in model 5?

Discussion

- Pag. 21, the following sentence: “The results from the only study found in the literature that compared the LDL-C levels estimated using the two equations in an adult population cannot be compared with the results of the present study because the objectives were different, considering that no comparisons were made between the two equations but, rather, between their results and the direct method” suggests me a question to the authors: Why you did not measure directly the LDL-C or why you did not discuss the two calculated values on the light of papers which have done this comparison? (in Pag. 24 this was considered a limitation of the paper: “Another limitation is the fact that the LDL-C values were not assessed by the direct method”);

- Pag. 21, the following sentence; “MDA was the only variable that positively influenced LDL-C concentrations, calculated based on the Friedewald (36) and Martin et al. “ but are you sure that MDA influence LDL-C and not vice versa?

- Pag. 22, there are different sentences suggesting a relationship between DNA methylation of some genes and the levels of LDL-C or some data like obesity, fat storages etc. Sorry, I am not an expert of the topic, but I want pose two questions: i) the epigenetic effects would be at tissue level, thus can you measure methylation at blood level? ii) methylation can have different effects according to the gene site in which occurred, can you consider that?

- Pag. 23, there is a sentence: “the population presented high MDA values, these values likely affect the lipid profile, increasing the LDL-C concentrations in the analysed models in the study population, with 75% to 67% of the LDL-C concentrations considered adequate, according to the two equations used.” But, again, MDA is more likely affected by lipid profile and perhaps by micro-inflammatory conditions which occur when there is obesity and high LDL-C;

- Pag. 23, “…several studies have shown that MDA and LDL-C values decrease in animal models with interventions based on the intake of different foods (21, 22, 27).” First of all it would be useful to better explain which foods are decreasing the LDL-C, moreover I am asking why very few or no dietary data of the experiment are been discussed;

- Pag. 24, in fact, in the Pag. 24, some sentences are concerned with the above topic: “…here was a positive influence of age, habitual cholesterol intake and WC on LDL-C levels.” Then “…even when the total fat intake was within the reference range (29%) (70) and the habitual cholesterol intake was 243.51 mg, an influence of this intake on LDL-C levels was identified. The explanation for this discrepancy may involve the type of fat consumed.” These topics are very important for a better explanation of the results and therefore the authors are invited to show the results of the dietary survey and to discuss them.

Conclusions

Must be changed according to the suggested improvements of the discussion.

6. PLOS authors have the option to publish the peer review history of their article (what does this mean?). If published, this will include your full peer review and any attached files.

Reviewer #1: No

---

## [Author Response · Author response to Decision Letter 0]

30 Jun 2020

A rebuttal letter that answers each point raised by the academic editor and reviewer. 

Journal Requirements:

2. Our internal editors have evaluated your manuscript and determined that it is within the scope of our 'Primary and Secondary Prevention of Cardiovascular Disease' Call for Papers. This collection of papers is headed by a team of Guest Editors for PLOS ONE and will encompass a diverse range of research articles. Additional information can be found on our announcement page: (https://collections.plos.org/s/prevention-cardiovascular). If you would like your manuscript to be considered for this collection, please let us know in your cover letter and we will ensure that your paper is treated as if you were responding to this call. If you would prefer to remove your manuscript from collection consideration, please specify this in the cover letter.

'We thank the funding agencies that made the project possible: Brazilian National Council for

Scientific and Technological Development (Conselho Nacional Desenvolvimento Científico e

Tecnológico - CNPq), Ministry of Health, and Research Support Foundation of the State of

Paraíba (Fundação de Amparo à Pesquisa do Estado da Paraíba - FAPESQ); and the research

collaborators (Graduate Program in Nutritional Sciences, University of Paraíba).'

Answer: Withdrawn.

'NO - Include this sentence at the end of your statement: The funders had no role in

study design, data collection and analysis, decision to publish, or preparation of the

manuscript.'

Answer: The database of the present study is present in the public repository, and can be viewed through the link: https://osf.io/t6zn8/?view_only=026188923f00464b8ab252bbc74ef129

Answer: Jéssica Vicky Bernardo de Oliveira ; ORCID: 0000-0002-4912-6849

Additional Editor Comments (if provided):

Reviewers' comments:

Reviewer's Responses to Questions

Comments to the Author

1. Is the manuscript technically sound, and do the data support the conclusions?

Reviewer #1: Partly

2. Has the statistical analysis been performed appropriately and rigorously?

Reviewer #1: N/A

3. Have the authors made all data underlying the findings in their manuscript fully available?

Reviewer #1: Yes

4. Is the manuscript presented in an intelligible fashion and written in standard English?

Reviewer #1: Yes 

5. Review Comments to the Author

Reviewer #1: The paper “The direct correlation between oxidative stress and LDL-C levels in adults is maintained by the Friedewald and Martin equations, but the methylation levels in the MTHFR and ADRB3 genes differ” by Jéssica Vicky Bernardo de Oliveira et al. is of some interest but I have some doubts which are difficult to be showed because the text is without pages and lines number. Therefore, I have showed the words-sentences and some comments have been done.

The number of pages is considering the whole document I received (sorry, but next time I shall not make any referee work on similar situation).

Introduction:

- Pag. 8, the sentence: “The biochemical, anthropometric and consumption variables are justified based on the following description:“ does not seem to be complete.

 Answer: Was withdrawn.

Materials and Methods

- Pag. 11, …” Weight and height measurements were carried out in triplicate…” did they occurred at the 3 times of dietary recalls? Please to explain;

Answer: Weight and height measurements were taken three times in the same day, so that there was no error.

- Pag. 13, it is suggested: “blood samples were collected at 8:30 a.m. ± 10 min from all individuals to avoid a potential sampling effect.” Please to explain the reason.

Answer: It is known that the gene expression of nucleated blood cells changes ex-vivo shortly after collection [1]. In this sense, even though the DNA suffers less interference from pre-analytical variables, because it has the nitrogen bases more protected in the double helix, with a minimum space for enzymatic attacks [2], a schedule that allows the shortest time between collection was standardized, transport and storage of samples.

[1] Malentacchi F, Pazzagli M, Simi L, Orlando C, Wyrich R, Gunther K et al. SPIDIA-RNA: second external quality assessment for the pre-analytical phase of blood samples used for RNA based analyses. PLoS One. 2014; 9:e112293.

[2] Ziehler WA, Engelke DR. Probing RNA structure with chemical reagentes and enzymes. Curr Protoc Nucleic Acid Chem 2001; Chapter 6:Unit 6 1.

Results

- Pag. 18, Table 2 I cannot imagine that Homocysteine could be “mol/L”

Answer: The typo was corrected. µmol / L

- Pag. 19, Table 3 only dietary cholesterol has been included? See some notes in Portuguese

Answer: The habitual consumption variables added in model 3 of the statistics were: CHO, PTN, LIP, vitamin A, beta-carotene, vitamin C, vitamin E, vitamin B12, polyunsaturated fats, monounsaturated fats, oleic acid, omega 3, omega 6, saturated fat, cholesterol, trans fats, selenium and folate, however, only the values of the variable (cholesterol) that showed a significant association with LDL-C values were added in Table 3.

Table 1 shows the blood cholesterol data.

The Portuguese phrase was translated into English.

- Pag. 20, Table 4 the LPL gene was not included in model 5?

Answer: It has been added.

Discussion

- Pag. 21, the following sentence: “The results from the only study found in the literature that compared the LDL-C levels estimated using the two equations in an adult population cannot be compared with the results of the present study because the objectives were different, considering that no comparisons were made between the two equations but, rather, between their results and the direct method” suggests me a question to the authors: Why you did not measure directly the LDL-C or why you did not discuss the two calculated values on the light of papers which have done this comparison? (in Pag. 24 this was considered a limitation of the paper: “Another limitation is the fact that the LDL-C values were not assessed by the direct method”);

Answer: Although it is important to accurately assess LDL-C through direct measurements, in routine clinical practice, LDL-C levels are generally estimated worldwide using the Friedewald formula [3], when triglyceride levels are below 400 mg / dL. The high cost of analysis of direct measurement methods (such as homogeneous enzyme assay or use of ultracentrifuge) made the use in the present population-based study unfeasible.

3. Catapano AL, Graham I, de Backer G, Wiklund O, Chapman MJ, Drexel H, et al. 2016 ESC/EAS Guidelines for the Management of Dyslipidaemias. Rev Española Cardiol. 2017;70(2):115.

Zabłocka-Słowińska, k. Oxidative stress in lung cancer patients is associated with altered serum markers of lipid metabolism. PLOS ONE, 2019. Doi: 10.1371/journal.pone.0215246

The results of the study by Lee et al. (14) are presented according to the agreement with the direct method, which makes the comparison unfeasible. Lee et al. (14) observed that the Martin equation exhibited significantly higher overall agreement with the classifications of the NCEP-ATP III guidelines, than the Friedewald equation, as already described in the present article.

14. Lee J, Jang S, Son H. Validation of the Martin Method for Estimating Low-Density Lipoprotein Cholesterol Levels in Korean Adults: Findings from the Korea National Health and Nutrition Examination Survey, 2009-2011. Passi AG, editor. PLoS One. 2016;11(1):1–14. 

- Pag. 21, the following sentence; “MDA was the only variable that positively influenced LDL-C concentrations, calculated based on the Friedewald (36) and Martin et al. “ but are you sure that MDA influence LDL-C and not vice versa?

In the study by Yang et al. [23] it was demonstrated that the impairment of the integrity and survival of human coronary artery endothelial cells (HCAECs) induced by oxidized LDL is mediated by the MDA epitope and involves the methylation of the prosurvival fibroblast growth factor 2 (FGF2) promoter and the negative regulation of intracellular FGF2. Signaling pathway through which MDA can mediate LDL-C oxidation.

23. Yang TC, Chen YJ, Chang SF, Chen CH, Chang PY, Lu SC. Malondialdehyde mediates oxidized LDL-induced coronary toxicity through the Akt-FGF2 pathway via DNA methylation. J Biomed Sci. 2014;21(1):1–12. 

According to the reading of the scientific community, we observed that there is no consensus regarding the questioned influence. However, in the present study we identified that the MDA was positively related to the LDL-C values.

- Pag. 22, there are different sentences suggesting a relationship between DNA methylation of some genes and the levels of LDL-C or some data like obesity, fat storages etc. Sorry, I am not an expert of the topic, but I want pose two questions: i) the epigenetic effects would be at tissue level, thus can you measure methylation at blood level? ii) methylation can have different effects according to the gene site in which occurred, can you consider that?

Answer: We fully understand your questioning, which was justified in our published article.

Lima, R.P.A.; NASCIMENTO, R.A.F. Effect of a diet containing folate and hazelnut oil capsule on the methylation level of the ADRB3 gene, lipid profile and oxidative stress in overweight or obese women, Clinical Epigenetics, 2017. DOI 10.1186/s13148-017-0407-6.

The blood was chosen for the analysis as it is a metabolically active tissue, with an important role in the adverse inflammatory and vascular consequences of adiposity, and is widely used for clinical diagnostic purposes.

- Pag. 23, there is a sentence: “the population presented high MDA values, these values likely affect the lipid profile, increasing the LDL-C concentrations in the analysed models in the study population, with 75% to 67% of the LDL-C concentrations considered adequate, according to the two equations used.” But, again, MDA is more likely affected by lipid profile and perhaps by micro-inflammatory conditions which occur when there is obesity and high LDL-C;

Answers: As suggested, changes were made to the text.

Malondialdehyde (MDA) is a final product of oxidative decomposition initiated by radical polyunsaturated fatty acids; therefore, it is often used as a biomarker of oxidative stress (23).

23. Yang TC, Chen YJ, Chang SF, Chen CH, Chang PY, Lu SC. Malondialdehyde mediates oxidized LDL-induced coronary toxicity through the Akt-FGF2 pathway via DNA methylation. J Biomed Sci. 2014;21(1):1–12.

[59] Lopes-Virella MF, Hunt KJ, Baker NL, Virella G, Moritz T. The levels of MDA-LDL in circulating immune complexes predict myocardial infarction in the VADT study. Atherosclerosis. 2012;224(2):526–31.

- Pag. 23, “…several studies have shown that MDA and LDL-C values decrease in animal models with interventions based on the intake of different foods (21, 22, 27).” First of all it would be useful to better explain which foods are decreasing the LDL-C, moreover I am asking why very few or no dietary data of the experiment are been discussed;

Answers: Regarding the relationship between MDA and LDL-C, several studies have shown that MDA and LDL-C values decrease in animal models with interventions based on the intake of different foods: Garlic (Allium sativum L.) polysaccharide (21), Chicory (Cichorium intybus L.) polysaccharides (22), Morechella esculenta polysaccharide (27).

As for dietary consumption, the authors were concerned about the extent of the discussion. If necessary, we add.

21. Wang Y, Guan M, Zhao X, Li X. Effects of garlic polysaccharide on alcoholic liver fibrosis and intestinal microflora in mice. Pharm Biol. 2018;56(1):325–32. 

22. Wu Y, Zhou F, Jiang H, Wang Z, Hua C, Zhang Y. Chicory (Cichorium intybus L.) polysaccharides attenuate high-fat diet induced non-alcoholic fatty liver disease via AMPK activation. Int J Biol Macromol. 2018;118:886–95. 

27. Dong Y, Qi Y, Liu M, Song X, Zhang C, Jiao X, et al. Antioxidant, anti-hyperlipidemia Land hepatic protection of enzyme-assisted Morehella esculenta polysaccharide. Int J Biol Macromol. 2018;120(2018):1490–9.

- Pag. 24, in fact, in the Pag. 24, some sentences are concerned with the above topic: “…here was a positive influence of age, habitual cholesterol intake and WC on LDL-C levels.” Then “…even when the total fat intake was within the reference range (29%) (70) and the habitual cholesterol intake was 243.51 mg, an influence of this intake on LDL-C levels was identified. The explanation for this discrepancy may involve the type of fat consumed.” These topics are very important for a better explanation of the results and therefore the authors are invited to show the results of the dietary survey and to discuss them

Answer: We added to the discussion as requested, however, we did not add the tables due to the length of the article, but if it is convenient for the reviewers, we will add it.

Another explanation for this discrepancy may involve the type of fat consumed, since, in the case of the present study, the consumption of saturated fatty acids was 12%, exceeding the recommendation according to the National Institute of Heart, Lung and Blood: Detection, assessment and treatment of high blood cholesterol levels in adults (adult treatment panel III), final report, US Department of Health and Human Services, NIH Publication No. 02-5215, Bethesda, Maryland, September 2002.

Conclusions

Must be changed according to the suggested improvements of the discussion.

6. PLOS authors have the option to publish the peer review history of their article (what does this mean?). If published, this will include your full peer review and any attached files.

Do you want your identity to be public for this peer review? For information about this choice, including consent withdrawal, please see our Privacy Policy.

Reviewer #1: No

---

## [Decision Letter · Decision Letter 1]

5 Aug 2020

PONE-D-20-07249R1

The direct correlation between oxidative stress and LDL-C levels in adults is maintained by the Friedewald and Martin equations, but the methylation levels in the MTHFR and ADRB3 genes differ

PLOS ONE

Dear Dr. Oliveira,

Thank you for submitting your manuscript to PLOS ONE. After careful consideration, we feel that it has merit but does not fully meet PLOS ONE’s publication criteria as it currently stands. Therefore, we invite you to submit a revised version of the manuscript that addresses the points raised during the review process.

We look forward to receiving your revised manuscript.

Kind regards,

Juan J Loor

Academic Editor

PLOS ONE

Reviewers' comments:

Reviewer's Responses to Questions

**Comments to the Author**

1. If the authors have adequately addressed your comments raised in a previous round of review and you feel that this manuscript is now acceptable for publication, you may indicate that here to bypass the “Comments to the Author” section, enter your conflict of interest statement in the “Confidential to Editor” section, and submit your "Accept" recommendation.

Reviewer #1: (No Response)

2. Is the manuscript technically sound, and do the data support the conclusions?

Reviewer #1: Yes

3. Has the statistical analysis been performed appropriately and rigorously? 

Reviewer #1: Yes

4. Have the authors made all data underlying the findings in their manuscript fully available?

Reviewer #1: Yes

5. Is the manuscript presented in an intelligible fashion and written in standard English?

Reviewer #1: (No Response)

6. Review Comments to the Author

Reviewer #1: The paper “The direct correlation between oxidative stress and LDL-C levels in adults is maintained by the Friedewald and Martin equations, but the methylation levels in the MTHFR and ADRB3 genes differ” by Jéssica Vicky Bernardo de Oliveira et al. has been modified according previous suggestions and now can be reviewed more easily.

With respect to Introduction, Materials and Methods and Results, the changes are satisfactorily (except in Table 1: Triglycerides (mg/d); while for Discussion, some further changes would be useful:

- Lines 410-414, perhaps the authors could spend some words to talk about the relationship between direct measurement and calculated values of LDL-C observed by Lee et al.;

- Lines 478-481 and later line 488 are concerning oxidative stress and inflammation which are often related; in this case the inflammation (CRP) seems at low levels therefore I ask whether the evaluated diets could suggest a lack of antioxidants (as suggested in line 496);

- Line 499 and till 519, the levels of triglycerides, that of course are carried out by LDL, are affected by several factors and some are dietary (not only total fat, SAFA etc., but also the high levels of carbohydrates; can the authors better show the results of their dietary survey and whether some nutrient levels are related to LDL-C, MDA etc.?

7. PLOS authors have the option to publish the peer review history of their article (what does this mean?). If published, this will include your full peer review and any attached files.

Reviewer #1: No

---

## [Author Response · Author response to Decision Letter 1]

10 Aug 2020

- Pag. 8, the sentence: “The biochemical, anthropometric and consumption variables are justified based on the following description:“ does not seem to be complete.

 Answer: Phrase withdrawn

Materials and Methods

- Pag. 11, …” Weight and height measurements were carried out in triplicate…” did they occurred at the 3 times of dietary recalls? Please to explain;

Answer: Weight and height measurements were taken three times in the same day, so that there was no error.

- Pag. 13, it is suggested: “blood samples were collected at 8:30 a.m. ± 10 min from all individuals to avoid a potential sampling effect.” Please to explain the reason.

Answer: It is known that the gene expression of nucleated blood cells changes ex-vivo shortly after collection [1]. In this sense, even though the DNA suffers less interference from pre-analytical variables, because it has the nitrogen bases more protected in the double helix, with a minimum space for enzymatic attacks [2], a schedule that allows the shortest time between collection was standardized, transport and storage of samples.

[1] Malentacchi F, Pazzagli M, Simi L, Orlando C, Wyrich R, Gunther K et al. SPIDIA-RNA: second external quality assessment for the pre-analytical phase of blood samples used for RNA based analyses. PLoS One. 2014; 9:e112293.

[2] Ziehler WA, Engelke DR. Probing RNA structure with chemical reagentes and enzymes. Curr Protoc Nucleic Acid Chem 2001; Chapter 6:Unit 6 1.

Results

- Pag. 18, Table 2 I cannot imagine that Homocysteine could be “mol/L”

Answer: The typo was corrected. µmol / L

- Pag. 19, Table 3 only dietary cholesterol has been included? See some notes in Portuguese

Answer: The habitual consumption variables added in model 3 of the statistics were: CHO, PTN, LIP, vitamin A, beta-carotene, vitamin C, vitamin E, vitamin B12, polyunsaturated fats, monounsaturated fats, oleic acid, omega 3, omega 6, saturated fat, cholesterol, trans fats, selenium and folate, however, only the values of the variable (cholesterol) that showed a significant association with LDL-C values were added in Table 3.

Table 1 shows the blood cholesterol data.

The Portuguese phrase was translated into English.

- Pag. 20, Table 4 the LPL gene was not included in model 5?

Answer: It has been added.

Discussion

- Pag. 21, the following sentence: “The results from the only study found in the literature that compared the LDL-C levels estimated using the two equations in an adult population cannot be compared with the results of the present study because the objectives were different, considering that no comparisons were made between the two equations but, rather, between their results and the direct method” suggests me a question to the authors: Why you did not measure directly the LDL-C or why you did not discuss the two calculated values on the light of papers which have done this comparison? (in Pag. 24 this was considered a limitation of the paper: “Another limitation is the fact that the LDL-C values were not assessed by the direct method”);

Answer: Although it is important to accurately assess LDL-C through direct measurements, in routine clinical practice, LDL-C levels are generally estimated worldwide using the Friedewald formula [3], when triglyceride levels are below 400 mg / dL. The high cost of analysis of direct measurement methods (such as homogeneous enzyme assay or use of ultracentrifuge) made the use in the present population-based study unfeasible.

3. Catapano AL, Graham I, de Backer G, Wiklund O, Chapman MJ, Drexel H, et al. 2016 ESC/EAS Guidelines for the Management of Dyslipidaemias. Rev Española Cardiol. 2017;70(2):115.

Zabłocka-Słowińska, k. Oxidative stress in lung cancer patients is associated with altered serum markers of lipid metabolism. PLOS ONE, 2019. Doi: 10.1371/journal.pone.0215246

The results of the study by Lee et al. (14) are presented according to the agreement with the direct method, which makes the comparison unfeasible. Lee et al. (14) observed that the Martin equation exhibited significantly higher overall agreement with the classifications of the NCEP-ATP III guidelines, than the Friedewald equation, as already described in the present article.

14. Lee J, Jang S, Son H. Validation of the Martin Method for Estimating Low-Density Lipoprotein Cholesterol Levels in Korean Adults: Findings from the Korea National Health and Nutrition Examination Survey, 2009-2011. Passi AG, editor. PLoS One. 2016;11(1):1–14. 

- Pag. 21, the following sentence; “MDA was the only variable that positively influenced LDL-C concentrations, calculated based on the Friedewald (36) and Martin et al. “ but are you sure that MDA influence LDL-C and not vice versa?

In the study by Yang et al. [23] it was demonstrated that the impairment of the integrity and survival of human coronary artery endothelial cells (HCAECs) induced by oxidized LDL is mediated by the MDA epitope and involves the methylation of the prosurvival fibroblast growth factor 2 (FGF2) promoter and the negative regulation of intracellular FGF2. Signaling pathway through which MDA can mediate LDL-C oxidation.

23. Yang TC, Chen YJ, Chang SF, Chen CH, Chang PY, Lu SC. Malondialdehyde mediates oxidized LDL-induced coronary toxicity through the Akt-FGF2 pathway via DNA methylation. J Biomed Sci. 2014;21(1):1–12. 

According to the reading of the scientific community, we observed that there is no consensus regarding the questioned influence. However, in the present study we identified that the MDA was positively related to the LDL-C values.

- Pag. 22, there are different sentences suggesting a relationship between DNA methylation of some genes and the levels of LDL-C or some data like obesity, fat storages etc. Sorry, I am not an expert of the topic, but I want pose two questions: i) the epigenetic effects would be at tissue level, thus can you measure methylation at blood level? ii) methylation can have different effects according to the gene site in which occurred, can you consider that?

Answer: We fully understand your questioning, which was justified in our published article.

Lima, R.P.A.; NASCIMENTO, R.A.F. Effect of a diet containing folate and hazelnut oil capsule on the methylation level of the ADRB3 gene, lipid profile and oxidative stress in overweight or obese women, Clinical Epigenetics, 2017. DOI 10.1186/s13148-017-0407-6.

The blood was chosen for the analysis as it is a metabolically active tissue, with an important role in the adverse inflammatory and vascular consequences of adiposity, and is widely used for clinical diagnostic purposes.

- Pag. 23, there is a sentence: “the population presented high MDA values, these values likely affect the lipid profile, increasing the LDL-C concentrations in the analysed models in the study population, with 75% to 67% of the LDL-C concentrations considered adequate, according to the two equations used.” But, again, MDA is more likely affected by lipid profile and perhaps by micro-inflammatory conditions which occur when there is obesity and high LDL-C;

Answers: As suggested, changes were made to the text.

Malondialdehyde (MDA) is a final product of oxidative decomposition initiated by radical polyunsaturated fatty acids; therefore, it is often used as a biomarker of oxidative stress (23).

23. Yang TC, Chen YJ, Chang SF, Chen CH, Chang PY, Lu SC. Malondialdehyde mediates oxidized LDL-induced coronary toxicity through the Akt-FGF2 pathway via DNA methylation. J Biomed Sci. 2014;21(1):1–12.

[59] Lopes-Virella MF, Hunt KJ, Baker NL, Virella G, Moritz T. The levels of MDA-LDL in circulating immune complexes predict myocardial infarction in the VADT study. Atherosclerosis. 2012;224(2):526–31.

- Pag. 23, “…several studies have shown that MDA and LDL-C values decrease in animal models with interventions based on the intake of different foods (21, 22, 27).” First of all it would be useful to better explain which foods are decreasing the LDL-C, moreover I am asking why very few or no dietary data of the experiment are been discussed;

Answers: Regarding the relationship between MDA and LDL-C, several studies have shown that MDA and LDL-C values decrease in animal models with interventions based on the intake of different foods: Garlic (Allium sativum L.) polysaccharide (21), Chicory (Cichorium intybus L.) polysaccharides (22), Morechella esculenta polysaccharide (27).

As for dietary consumption, the authors were concerned about the extent of the discussion. If necessary, we add.

21. Wang Y, Guan M, Zhao X, Li X. Effects of garlic polysaccharide on alcoholic liver fibrosis and intestinal microflora in mice. Pharm Biol. 2018;56(1):325–32. 

22. Wu Y, Zhou F, Jiang H, Wang Z, Hua C, Zhang Y. Chicory (Cichorium intybus L.) polysaccharides attenuate high-fat diet induced non-alcoholic fatty liver disease via AMPK activation. Int J Biol Macromol. 2018;118:886–95. 

27. Dong Y, Qi Y, Liu M, Song X, Zhang C, Jiao X, et al. Antioxidant, anti-hyperlipidemia Land hepatic protection of enzyme-assisted Morehella esculenta polysaccharide. Int J Biol Macromol. 2018;120(2018):1490–9.

- Pag. 24, in fact, in the Pag. 24, some sentences are concerned with the above topic: “…here was a positive influence of age, habitual cholesterol intake and WC on LDL-C levels.” Then “…even when the total fat intake was within the reference range (29%) (70) and the habitual cholesterol intake was 243.51 mg, an influence of this intake on LDL-C levels was identified. The explanation for this discrepancy may involve the type of fat consumed.” These topics are very important for a better explanation of the results and therefore the authors are invited to show the results of the dietary survey and to discuss them

Answer: We added to the discussion as requested, however, we did not add the tables due to the length of the article, but if it is convenient for the reviewers, we will add it.

Another explanation for this discrepancy may involve the type of fat consumed, since, in the case of the present study, the consumption of saturated fatty acids was 12%, exceeding the recommendation according to the National Institute of Heart, Lung and Blood: Detection, assessment and treatment of high blood cholesterol levels in adults (adult treatment panel III), final report, US Department of Health and Human Services, NIH Publication No. 02-5215, Bethesda, Maryland, September 2002.

---

## [Decision Letter · Decision Letter 2]

17 Sep 2020

The direct correlation between oxidative stress and LDL-C levels in adults is maintained by the Friedewald and Martin equations, but the methylation levels in the MTHFR and ADRB3 genes differ

PONE-D-20-07249R2

Dear Dr. Oliveira,

We’re pleased to inform you that your manuscript has been judged scientifically suitable for publication and will be formally accepted for publication once it meets all outstanding technical requirements.

Kind regards,

Juan J Loor

Academic Editor

PLOS ONE

Additional Editor Comments (optional):

Reviewers' comments:

Reviewer's Responses to Questions

**Comments to the Author**

1. If the authors have adequately addressed your comments raised in a previous round of review and you feel that this manuscript is now acceptable for publication, you may indicate that here to bypass the “Comments to the Author” section, enter your conflict of interest statement in the “Confidential to Editor” section, and submit your "Accept" recommendation.

Reviewer #1: All comments have been addressed

2. Is the manuscript technically sound, and do the data support the conclusions?

Reviewer #1: Yes

3. Has the statistical analysis been performed appropriately and rigorously? 

Reviewer #1: Yes

4. Have the authors made all data underlying the findings in their manuscript fully available?

Reviewer #1: Yes

5. Is the manuscript presented in an intelligible fashion and written in standard English?

Reviewer #1: Yes

6. Review Comments to the Author

Reviewer #1: Please to check the measure unit of triglyceride in Table 1, I am surprise to see mg/d; perhaps there is some mistake

7. PLOS authors have the option to publish the peer review history of their article (what does this mean?). If published, this will include your full peer review and any attached files.

Reviewer #1: No

---

## [Editor Report · Acceptance letter]

30 Oct 2020

PONE-D-20-07249R2 

The direct correlation between oxidative stress and LDL-C levels in adults is maintained by the Friedewald and Martin equations, but the methylation levels in the MTHFR and ADRB3 genes differ 

Dear Dr. Oliveira:

I'm pleased to inform you that your manuscript has been deemed suitable for publication in PLOS ONE. Congratulations! Your manuscript is now with our production department. 

Kind regards, 

on behalf of

Dr. Juan J Loor 

Academic Editor

PLOS ONE